# Clinical Effect of the C-Reactive Protein to Serum Albumin Ratio in Patients with Metastatic Gastric or Gastroesophageal Junction Cancer Treated with Trifluridine/Tipiracil

**DOI:** 10.3390/jpm13060923

**Published:** 2023-05-31

**Authors:** Itaru Hashimoto, Kazuki Kano, Shizune Onuma, Hideaki Suematsu, Shinsuke Nagasawa, Kyohei Kanematsu, Kyoko Furusawa, Tomomi Hamaguchi, Mamoru Watanabe, Kei Hayashi, Mitsuhiro Furuta, Yasuhiro Inokuchi, Nozomu Machida, Toru Aoyama, Takanobu Yamada, Yasushi Rino, Takashi Ogata, Takashi Oshima

**Affiliations:** 1Department of Gastrointestinal Surgery, Kanagawa Cancer Center, Yokohama 241-8515, Kanagawa, Japan; 2Department of Surgery, Yokohama City University, Yokohama 236-0004, Kanagawa, Japan; 3Department of Gastroenterology, Kanagawa Cancer Center, Yokohama 241-8515, Kanagawa, Japan

**Keywords:** C-reactive protein to serum albumin ratio, gastric cancer, gastroesophageal junction cancer, trifluridine/tipiracil

## Abstract

Trifluridine/tipiracil (FTD/TPI) is an oral anticancer agent used as a third- or later-line treatment for patients with metastatic gastric cancer/gastroesophageal junction cancer (mGC/GEJC). The C-reactive protein-to-serum albumin ratio (CAR) is an inflammation-based prognostic marker in gastric cancer. This retrospective study evaluated CAR’s clinical significance as a prognostic factor in 64 patients with mGC/GEJC administered FTD/TPI as a third- or later-line therapy. Patients were categorized into high- and low-CAR groups based on pre-treatment blood data. This study evaluated associations between CAR and overall survival (OS), progression-free survival (PFS), clinicopathological features, treatment efficacy, and adverse events. The high-CAR group had significantly worse Eastern Cooperative Oncology Group performance status, a higher prevalence of patients administered with a single course of FTD/TPI, and a higher rate of patients not administered chemotherapy after FTD/TPI therapy than the low-CAR group. Median OS and PFS were significantly poorer in the high-CAR group than in the low-CAR group (113 vs. 399 days; *p* < 0.001 and 39 vs. 112 days; *p* < 0.001, respectively). In multivariate analysis, high CAR was an independent prognostic factor for OS and PFS. The overall response rate was not significantly different between the high- and low-CAR groups. Regarding adverse events, the high-CAR group had a significantly lower incidence of neutropenia and a higher incidence of fatigue than the low-CAR group. Therefore, CAR may be a potentially useful prognostic factor for patients with mGC/GEJC treated with FTD/TPI as third- or later-line chemotherapy.

## 1. Introduction

Gastric cancer (GC) is the fifth and fourth leading cause of cancer and cancer-related mortality, respectively [1]. Despite the recent advances in diagnosis, surgery, and systemic chemotherapy, treatment outcomes for GC remain unsatisfactory and require further improvement [2]. Recently, innovative systemic chemotherapy regimens for metastatic gastric cancer/gastroesophageal junction cancer (mGC/GEJC) have been developed, including nivolumab [3,4]; pembrolizumab [5], based on microsatellite instability status; and trastuzumab deruxtecan, based on human epidermal growth factor receptor type 2 status [6].

Furthermore, trifluridine/tipiracil (FTD/TPI) is an oral anticancer agent comprising a thymidine-based nucleoside analog, FTD, and a thymidine phosphorylase (TP) inhibitor, TPI. The underlying mechanism of action of FTD/TPI is characterized by FTD substitution of thymidine in DNA, leading to impaired DNA function [7,8,9] and the prevention of FTD degradation by TPI, thereby augmenting the bioavailability of FTD [10]. In GC, the results of the TAGS study demonstrated the efficacy of FTD/TPI therapy in treating heavily pre-treated patients with mGC [11], thereby establishing it as a standard third-line chemotherapy option [12]. Several subgroup analyses have demonstrated the efficacy and tolerability of FTD/TPI in patients with gastrectomy [13] or GEJC [14] and older patients [15]. However, prognostic factors for patients with mGC/GEJC administered FTD/TPI treatment have not been thoroughly investigated.

Peri-treatment inflammatory and nutritional status are attracting attention as prognostic factors for patients with cancer [16,17]. Several studies have shown that the C-reactive protein/serum albumin ratio (CAR), which reflects the host’s inflammatory and nutritional status, is a useful prognostic marker in pre-treatment blood testing [18,19]. However, the quest to identify prognostic factors in patients with mGC/GEJC treated with FTD/TPI remain insufficient.

Therefore, this study aimed to evaluate CAR’s usefulness before initiating FTD/TPI therapy as one of the prognostic factors in patients with mGC/EGJC treated with FTD/TPI as a third- or later-line chemotherapy.

## 2. Materials and Methods

### 2.1. Ethical Approval

The Ethics Committee of Kanagawa Cancer Center approved all study protocols (approval number: 2022epidemiologic study-108), and all procedures were conducted following the Declaration of Helsinki of 1996.

### 2.2. Patients

In this retrospective study, we analyzed patients with mGC/GEJC treated with FTD/TPI chemotherapy at the Kanagawa Cancer Center between October 2019 and June 2022. The inclusion criteria were histological confirmation of adenocarcinoma, unresectable and metastatic disease, Eastern Cooperative Oncology Group performance status (ECOG PS) of 0–2, and prior treatment with FTD/TPI. Patients with a history of any other cancer that required aggressive treatment were excluded.

### 2.3. Assessment of Treatment Response and Adverse Events after FTD/TPI Therapy

The patients received oral FTD/TPI therapy (35 mg/m^2^ twice daily on days 1–5 and 8–12 every 28 days) [11] until disease progression, unacceptable adverse events, or patient refusal. Additionally, the treatment response was evaluated using tomography scans and the Response Evaluation Criteria in Solid Tumours [20]. Adverse events were assessed using the National Cancer Institute Common Terminology Criteria for Adverse Events.

### 2.4. Definition of CAR

Clinical blood samples from patients were typically transported directly to the blood draw room for immediate analysis. Blood samples were used to measure blood counts, biochemistry, coagulation factors, and tumor markers. We defined CAR using blood test data before initiating the first FTD/TPI cycle as follows:

CAR = C-reactive protein (mg/dL)/serum albumin (g/dL).

The cutoff value, defined using a receiver operating characteristic curve analysis for survival and death, was 0.09 (Figure 1). The patients were categorized into the high and low CAR groups.

### 2.5. Statistical Analyses

Categorical variables were compared using the χ^2^ or Fisher’s exact test as appropriate. Overall survival (OS) and progression-free survival (PFS) rates after initiating FTD/TPI therapy were evaluated using the Kaplan–Meier method and log-rank test. Furthermore, variables identified as significant (*p* < 0.05) in univariate analysis were considered candidates for multivariate COX regression analysis, and the results were presented as hazard ratios (HRs) with 95% confidence intervals (CIs). Statistical significance was set at *p* < 0.05. All statistical analyses were performed using the EZR (Saitama Medical Center, Jichi Medical University, Saitama, Japan), which is a graphical user interface for R (The R Foundation for Statistical Computing, Vienna, Austria).

## 3. Results

### 3.1. Patients

Overall, 64 participants were included in this study. The median age was 68 years (range, 35–85 years), the median number of treatment lines completed was 2 (range, 1–16), and the OS duration was 184 days (range, 18–970). There were 37 and 27 patients in the high and low CAR groups, respectively, using the cutoff point for the CAR.

### 3.2. Relationship between CAR and Clinicopathological Factors

The high-CAR group had a significantly worse Eastern Cooperative Oncology Group performance status (*p* = 0.022), a higher prevalence of patients receiving only a single course of FTD/TPI (*p* = 0.018), and a higher rate of patients who could not be administered any post-FTD/TPI therapy (*p* = 0.004) than the low-CAR group. However, no significant relationships were observed among age (*p* = 0.797), sex (*p* = 0.600), histological type (*p* = 0.621), human epidermal growth factor receptor 2 status (*p* = 1.000), macroscopic classification (*p* = 0.058), primary tumor site (*p* = 1.000), peritoneal metastasis (*p* = 0.129), hematogenous metastasis (*p* = 0.613), lymphatic metastasis (*p* = 0.802), number of metastatic sites (*p* = 0.454), number of previous regimens (*p* = 0.551), and previous immune checkpoint inhibitor administered (*p* = 1.000) between the high- and low-CAR groups (Table 1).

### 3.3. Median OS and PFS Based on the CAR in Patients with mGC/GEJC Treated with FTD/TPI

The median OS of patients with high CAR was significantly poorer than that of those with low CAR (113 days vs. 399 days; *p* < 0.001) (Figure 2). Additionally, the median PFS of patients with high CAR was significantly poorer than that of those with low CAR (39 days vs. 112 days; *p* < 0.001) (Figure 3).

### 3.4. Univariate and Multivariate Analyses of OS and PFS

Table 2 shows the results of univariate and multivariate analyses of OS in patients with GC who underwent FTD/TPI. In univariate analyses, ECOG PS (HR, 1.83; 95% CI, 1.02–3.29; *p* = 0.04), peritoneal metastasis (HR, 2.57; 95% CI, 1.37–4.82; *p* = 0.003), and CAR (HR, 4.48; 95% CI, 2.24–8.97; *p* < 0.001) were prognostic factors for OS. In multivariate analyses, CAR (HR, 3.56; 95% CI, 1.67–7.55; *p* < 0.001) was an independent prognostic factor for OS (Table 2). Table 3 shows the results of univariate and multivariate analyses of PFS in patients with GC who underwent FTD/TPI. In univariate analyses, peritoneal metastasis (HR, 1.78; 95% CI, 1.04–3.05; *p* = 0.04) and CAR (HR, 2.67; 95% CI, 1.56–4.66; *p* < 0.001) were prognostic factors for PFS. However, CAR (HR, 2.42; 95% CI, 1.33–4.41; *p* = 0.004) was an independent prognostic factor for PFS in multivariate analyses (Table 3).

### 3.5. Relationship between CAR and Treatment Response to FTD/TPI Therapy

The rates of complete response were 0.0% (*n* = 0) and 7.4% (*n* = 2), partial response rates were 5.4% (*n* = 2) and 14.8% (*n* = 4), stable disease rates were 21.6% (*n* = 8) and 29.6% (*n* = 8), progressive disease rates were 67.6% (*n* = 25) and 44.4% (*n* = 12), and not evaluated rates were 5.4% (*n* = 2) and 3.7% (*n* = 1) in the high- and low-CAR groups, respectively. The overall response rates (ORR) were 5.4% (*n* = 2) and 22.2% (*n* = 6) in the high- and low-CAR groups, respectively, which were superior to the high-CAR group but not significantly different (Table 4). Furthermore, the disease control rates (DCRs) were 27.0% (*n* = 10) and 51.9% (*n* = 14) in the high- and low-CAR groups, respectively, which were superior to the high-CAR group but not significantly different (Table 4).

### 3.6. Adverse Events of FTD/TPI Therapy in the High- and Low-CAR Groups

The high-CAR group had a significantly lower and higher incidence of neutropenia (*p* = 0.008) and fatigue (*p* = 0.008), respectively, than the low-CAR group (Table 5). However, no significant relationships were observed with total adverse events (*p* = 1.000), anorexia (*p* = 0.440), diarrhea (*p* = 0.632), decreased platelet count (*p* = 1.000), anemia (*p* = 1.000), biliary tract infection (*p* = 1.000), febrile neutropenia (*p* = 0.422), hyponatremia (*p* = 0.422), myalgia (*p* = 1.000), and peripheral sensory neuropathy (*p* = 1.000) between the high- and low-CAR groups.

## 4. Discussion

This study evaluated the clinical effect of CAR on survival outcomes in patients with mGC/EGJC treated with FTD/TPI therapy as a third- or later-line chemotherapy. The treatment responses, including ORR and DCR, were not significantly different between the high- and low-CAR groups. In the survival analysis, the OS and PFS were significantly worse in the high-CAR group than in the low-CAR group. However, in the multivariate analysis, CAR was an independent prognostic factor for OS and PFS. Furthermore, the high-CAR group had a higher and a lower incidence of fatigue and neutropenia, respectively, than the low-CAR group as an adverse effect of FTD/TPI.

Based on the TAGS trial results [11], FTD/TPI has become a novel chemotherapy regimen for mGC/GEJC following third-line treatment. To date, several reports assessing the efficacy and tolerability of patients with mGC/EGJC treated with FTD/TPI are available. Subgroup analyses of a randomized clinical trial focusing on previous gastrectomy showed that in the subgroup of patients who underwent gastrectomy, the HRs for OS and PFS in the group treated with FTD/TPI versus the placebo group were 0.57 (95% CI, 0.41–0.79) and 0.48 (95% CI, 0.35–0.65), respectively. However, in the subgroup of patients who did not undergo gastrectomy, the HRs for OS and PFS in the group treated with FTD/TPI versus the placebo group were 0.80 (95% CI, 0.60–1.06) and 0.65 (95% CI, 0.49–0.85), respectively [13]. Other subgroup analyses focusing on mEGJC showed that in the subgroup of patients with mEGJC, the HRs for OS and PFS in the group treated with FTD/TPI versus the placebo group were 0.75 (95% CI, 0.50–1.11) and 0.60 (95% CI, 0.41–0.88), respectively. In the subgroup of patients with mGC, the HRs for OS and PFS in the group treated with FTD/TPI versus the placebo group were 0.67 (95% CI, 0.52–0.87) and 0.59 (95% CI, 0.46–0.75), respectively [14]. Additionally, subgroup analyses focusing on age (<65, ≥65, and ≥75 years) showed that the HRs for OS in the FTD/TPI group versus placebo were 0.67 (95% CI, 0.51–0.89), 0.73 (95% CI, 0.52–1.02), and 0.67 (95% CI, 0.33–1.37), and the HRs for PFS were 0.68 (95% CI, 0.51–0.89), 0.44 (95% CI, 0.32–0.61), and 0.71 (95% CI, 0.37–1.36) in patients aged <65, ≥65, and ≥75 years, respectively [15].

Studies investigating prognostic factors for patients with mGC/GEJC treated with FTD/TPI therapy are limited. Nevertheless, FTD/TPI has been approved for treating metastatic colorectal cancer (mCRC), and several prognostic factors have been identified for patients with mCRC. A retrospective analysis among 47 patients receiving FTD/TPI or regorafenib as a last-line treatment showed that median OS was significantly shorter in those with KRAS-mutation type than in those with KRAS-wild type (154 vs. 223.5 days, *p* = 0.05). In multivariate analysis, KRAS-mutation type was an independent prognostic factor of OS (HR, 2.88; 95% CI, 1.09–4.02, *p* = 0.027) [21]. Additionally, a retrospective analysis among 14 patients treated with FTD/TPI showed that median OS was significantly longer in those treated for >18 months from the start of first-line treatment than in those treated for ≤18 months from the start of first-line therapy (7 months vs. 5 months, *p* = 0.029) [22]. Several reports have suggested that neutropenia during treatment with FTD/TPI is a useful prognostic factor in patients with mCRC. A retrospective study among 95 consecutive patients treated with FTD/TPI showed that grade 3–4 neutropenia was a significant predictive factor for PFS compared with grade 0–2 neutropenia in multivariate analysis (4.3 vs. 2.0 months; HR, 0.45; *p* = 0.01) [23]. Furthermore, a retrospective analysis of 14 patients treated with FTD/TPI revealed that the median OS was significantly longer in patients with severe neutropenia (grade 3–4) than in those without neutropenia and mild neutropenia (grade 1–2) (299 days vs. 184 days vs. 120 days, *p* = 0.045). In multivariate analysis, severe neutropenia (grade 3–4) was an independent prognostic factor of OS (HR, 0.442; 95% CI, 0.201–0.974, *p* = 0.042) [24]. Another retrospective analysis of 56 patients treated with FTD/TPI monotherapy or FTD/TPI plus bevacizumab demonstrated that the median OS and PFS were significantly higher in patients with chemotherapy-induced neutropenia (grade 3–4) than in those without neutropenia/mild neutropenia (grade 0–2) (*p* = 0.045 and *p* = 0.033, respectively) [25]. Moreover, several inflammatory and nutritional scores using peripheral blood tests have been reported as useful prognostic factors in patients with mCRC treated with FTD/TPI. A retrospective study among 40 patients treated with FTD/TPI showed that a ≥25% decrease in neutrophils was an independent prognostic factor for OS (HR, 0.28; 95% CI, 0.12–0.72, *p* = 0.01) [26]. A single institutional retrospective study among 33 patients with mCRC treated with FTD/TPI revealed that a high neutrophil-to-lymphocyte ratio (NLR) was an independent prognostic factor for PFS in multivariate analysis (HR, 6.26; 95% CI, 1.99–19.74, *p* = 0.002) [27]. Furthermore, a multi-center retrospective real-world analysis of 236 patients with mCRC treated with FTD/TPI showed that a low NLR was a favorable prognostic factor for OS in the multivariate analysis (HR, 0.56; 95% CI, 0.35–0.88, *p* = 0.01) [28]. However, a retrospective single-center study of 40 patients treated with FTD/TPI showed that the OS and PFS were significantly lower in the high-CAR group than in the low-CAR group (*p* <0.001 and *p* = 0.006, respectively). In multivariate analysis, high CAR was an independent prognostic factor for OS (HR, 6.48; 95% CI, 1.37–30.54, *p* = 0.018) [29].

Recently, inflammatory and nutritional markers have received increasing attention as prognostic predictors in patients with several gastrointestinal cancers [30,31]. Additionally, CAR is an indicator comprising C-reactive protein (CRP) as an inflammatory marker [32] and albumin as a reflection of nutritional status and fluid balance regulation [33]. Recent studies have reported that pre-treatment CRP, albumin, and CAR are useful prognostic factors for mGC/GEJC. A single-center retrospective study of 411 patients with unresectable or recurrent GC showed that the median survival was significantly shorter in those with high CAR than in those with low CAR (9.9 months vs. 14.8 months, *p* < 0.029) [18]. A multi-institutional cohort study of 278 patients with unresectable or recurrent GC treated with nivolumab showed that CRP levels ≤ 0.5 mg/dL (HR, 0.476, 95% CI 0.336–0.675, *p* < 0.001) and albumin levels >3.5 g/dL (HR, 0.688; 95% CI, 0.478–0.991, *p* = 0.045) were independently associated with improved outcomes in multivariate analysis [34]. Furthermore, a retrospective multi-center study of 97 patients with unresectable or recurrent GC who received nivolumab showed that the median OS was significantly shorter in the group with high changes in CAR from first-line chemotherapy (ΔCAR) than in the group with low ΔCAR (4.5 months vs. 9.4 months, *p* = 0.002). On multivariate analysis, low ΔCAR was an independent prognostic factor of OS (HR, 0.67; 95% CI, 0.35–0.91, *p* = 0.02) [19]. Considering the clinical application of CAR in patients with mGC/EGJC treated with FTD/TPI, using CAR as a prognostic predictor for patients with mGC/EGJC may be a factor in the decision to transition to the next line of treatment. In fact, in this study, the high-CAR group had a higher prevalence of patients receiving only a single course of FTD/TPI than the low-CAR group. This suggests that the high CAR group has not fully benefited from FTD/TPI treatment regarding disease status and adverse effects, which may contribute to poor prognosis. Therefore, accurately predicting prognosis is crucial before chemotherapy for patients with mGC/EGJC treated using FTD/TPI as a third- or later-line therapy with limited treatment options.

Although the mechanism through which CAR is useful as a prognostic factor for patients with mGC/EGJC treated with FTD/TPI therapy has not been elucidated, potential explanations related to the biological action process of FTD/TPI and cancer-related fatigue due to inflammation are available. The expression of TP and inflammation in cancer may play essential roles in the mechanism of action of FTD/TPI. Notably, TP serves as a limiting catalyst in the catabolism of thymidine [35,36] and plays a critical role in the angiogenesis and metabolism of anticancer drugs [35]. Additionally, TP-facilitated thymidine catabolism contributes to tumor viability under conditions of low nutrient status and the pathway for thymidine to the glycolytic cascade [36]. In GC, an immunohistochemical analysis (IHC) of 116 patients with GC showed that the TP expression in cancer-infiltrating inflammatory cells was associated with survival and lymph node metastasis [37]. Furthermore, a combined analysis of the blood test and IHC demonstrated that the systemic immune-inflammation index was related to TP-positive expression in 455 patients with GC who underwent curative surgery [38]. Therefore, assessing the degree of host inflammation using CAR may be useful as an indirect estimate of TP expression and a prognostic factor in patients with mGC/EGJC treated with FTD/TPI. From another perspective, systemic inflammation [39,40] is closely linked to cancer-related fatigue pre-, peri-, and post-cancer treatment [41], which is one of the most common side effects of FTD/TPI therapy [11]. Recently, several studies have shown that cancer-related fatigue may be associated with decreased survival due to cancer progression, treatment adherence, and poor quality of life [42,43,44]. In several types of cancer, CRP elevation during the peri-treatment period has been reportedly associated with fatigue [45,46,47]. In this study, the high-CAR group had significantly more fatigue than the low-CAR group (28.1% vs. 0%, *p* = 0.002), thereby supporting these previous reports on mGC/GEJC, and CAR may be an indirect predictor of fatigue and a beneficial prognostic factor in patients with mGC/EGJC treated with FTD/TPI.

This study had some limitations. First, a clinically relevant cutoff for the CAR ratio may require further investigation because there are relatively few reports on the CAR. Second, this was a single-cohort study, and the number of cases may have been too small to confirm CAR’s robustness as a beneficial prognostic factor. Therefore, multi-center studies with validation cohorts should validate CAR as a prognostic marker in patients with GC treated with FTD/TPI. Finally, this study did not compare the FTD/TPI group with the placebo group as was performed in the TAGS study cohort; therefore, further validation is needed to determine whether CAR is a true prognostic factor.

In conclusion, CAR may be a useful prognostic factor for patients with mGC/GEJC treated with FTD/TPI as third- or later-line chemotherapy regimens.

## Figures and Tables

**Figure 1 jpm-13-00923-f001:**
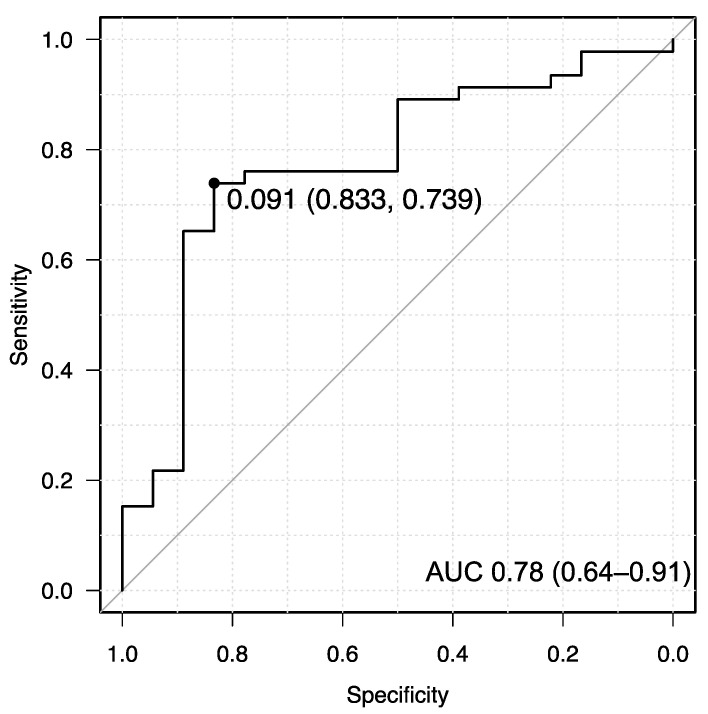
Definition of the cutoff value using receiver operating characteristic curve analysis on survival in CAR. AUC, area under the curve; CAR, C-reactive protein/serum albumin ratio.

**Figure 2 jpm-13-00923-f002:**
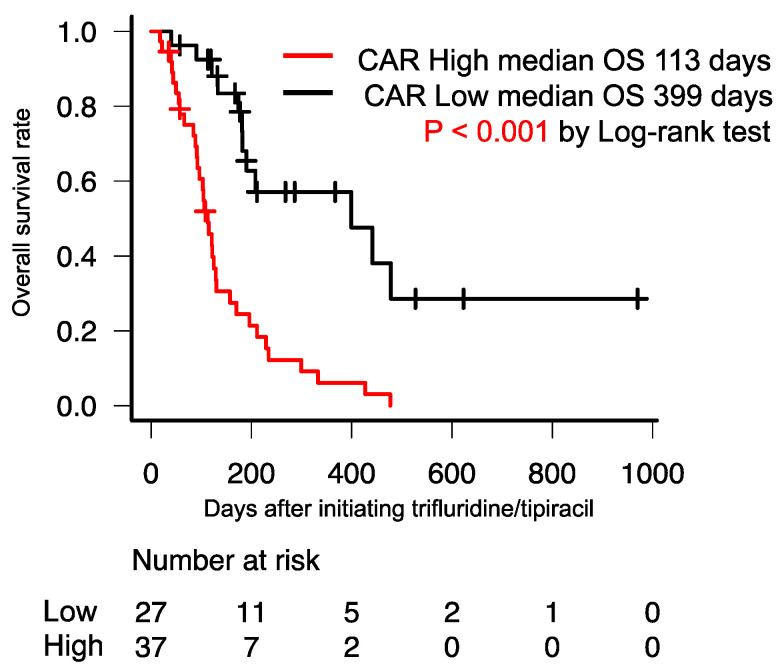
Kaplan–Meier curve for OS between the high- and low-CAR groups. The median OS of patients with high CAR was significantly shorter than that of those with low CAR (113 days vs. 399 days; *p* < 0.001). OS, overall survival; CAR, C-reactive protein-to-serum albumin ratio.

**Figure 3 jpm-13-00923-f003:**
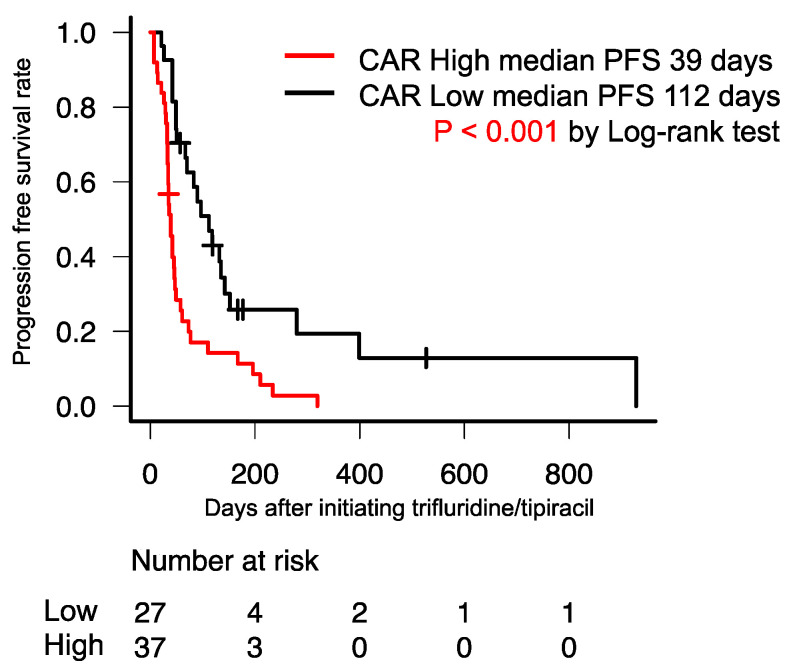
Kaplan–Meier curve for PFS between the high- and low-CAR groups. The median PFS of patients with high CAR was significantly poorer than that of those with low CAR (39 vs. 112 days; *p* < 0.001). CAR, C-reactive protein to serum albumin ratio; PFS, progression-free survival.

**Table 1 jpm-13-00923-t001:** Clinicopathological data between low- and high-CAR groups. The bold data indicate statistically significant (*p* < 0.05).

Variables	All Patients (*n* = 64)	CAR		*p*-Value
		Low (*n* = 27)	High (*n* = 37)	
Age	<65	10 (37.0)	16 (43.2)	0.797
	≧65	17 (63.0)	21 (56.8)	
Sex	Male	11 (40.7)	12 (32.4)	0.600
	Female	16 (59.3)	25 (67.6)	
ECOG PS	0	20 (74.1)	15 (40.5)	**0.022**
	1	6 (22.2)	19 (51.4)	
	2	1 (3.7)	3 (8.1)	
Histological type	Intestinal	12 (44.4)	19 (51.4)	0.621
	Diffuse	15 (55.6)	18 (48.6)	
HER2 status	Negative	24 (88.9)	33 (89.2)	1.000
	Positive	3 (11.1)	4 (10.8)	
Macroscopic classification	Non-type 4	25 (92.6)	27 (73.0)	0.058
	Type 4	2 (7.4)	10 (27.0)	
Primary tumor site	GC	21 (77.8)	29 (78.4)	1.000
	EGJC	6 (22.2)	8 (21.6)	
Disease status	Unresectable	12 (44.4)	25 (67.6)	0.078
	Recurrence	15 (55.6)	12 (32.4)	
Peritoneal metastasis	−	16 (59.3)	14 (37.8)	0.129
	+	11 (40.7)	23 (62.2)	
Hematogenous metastasis	−	12 (44.4)	20 (54.1)	0.613
	+	15 (55.6)	17 (45.9)	
Lymphatic metastasis	−	15 (55.6)	22 (59.5)	0.802
	+	12 (44.4)	15 (40.5)	
Number of metastatic sites	1	16 (59.3)	18 (48.6)	0.454
	≧2	11 (40.7)	19 (51.4)	
Number of previous regimens	2	20 (74.1)	30 (81.1)	0.551
	≧3	7 (25.9)	7 (18.9)	
Previous ICI administered	−	10 (37.0)	14 (37.8)	1.000
	+	17 (63.0)	23 (62.2)	
Number of FTD/TPI	1	5 (18.5)	18 (48.6)	**0.018**
	≧2	22 (81.5)	19 (51.4)	
Post-FTD/TPI therapy	−	12 (44.4)	30 (81.1)	**0.004**
	+	13 (48.1)	7 (18.9)	
	FTD/TPI ongoing	2 (7.4)	0 (0.0)	

CAR, C-reactive protein/serum albumin ratio; ECOG PS, Eastern Cooperative Oncology Group performance status; HER2, human epidermal growth factor receptor 2; GC, gastric cancer; GEJC: gastroesophageal junction cancer; FTD/TPI, trifluridine/tipiracil.; ICI, immune checkpoint inhibitor.

**Table 2 jpm-13-00923-t002:** Univariate and multivariate analyses of clinicopathological factors and CAR for OS. The bold data indicate statistically significant (*p* < 0.05).

Factors		Univariate		*p*-Value	Multivariate		*p*-Value
		HR	95% CI		HR	95% CI	
Age	<65	1					
	≧65	0.74	0.41–1.32	0.30			
Sex	Male	1					
	Female	0.62	0.34–1.13	0.12			
ECOG PS	0	1			1		
	1, 2	1.83	1.02–3.29	**0.04**	1.32	0.72–2.43	0.37
HER2 status	Negative	1					
	Positive	0.76	0.27–2.13	0.60			
Disease status	Recurrence	1					
	Unresectable	0.67	0.37–1.23	0.20			
Peritoneal metastasis	-	1			1		
	+	2.57	1.37–4.82	**0.003**	1.51	0.76–2.98	0.24
No. of previous regimens	3	1					
	≧4	0.90	0.43–1.87	0.77			
CAR	Low	1			1		
	High	4.48	2.24–8.97	**<0.001**	3.56	1.67–7.55	**<0.001**

HR, hazard ratio; CI, confidence interval; ECOG, Eastern Cooperative Oncology Group performance status; HER2, human epidermal growth factor receptor 2; CAR, C-reactive protein/serum albumin ratio.

**Table 3 jpm-13-00923-t003:** Univariate and multivariate analyses of clinicopathological factors and CAR for progression-free survival. The bold data indicate statistically significant (*p* < 0.05).

Factors		Univariate		*p*-Value	Multivariate		*p*-Value
		HR	95% CI		HR	95% CI	
Age	<65	1					
	≧65	0.68	0.40–1.16	0.16			
Sex	Male	1					
	Female	0.66	0.39–1.13	0.13			
ECOG PS	0	1					
	1, 2	1.56	0.92–2.65	0.10			
HER2 status	Negative	1					
	Positive	1.37	0.61–3.07	0.44			
Disease status	Recurrence	1					
	Unresectable	0.78	0.45–1.33	0.36			
Peritoneal metastasis	-	1			1		
	+	1.78	1.04–3.05	**0.04**	1.28	0.72–2.28	0.41
No. of previous regimens	3	1					
	≧4	0.80	0.42–1.51	0.49			
CAR	Low	1			1		
	High	2.67	1.53–4.66	**<0.001**	2.42	1.33–4.41	**0.004**

HR, hazard ratio; CI, confidence interval; HER2, human epidermal growth factor receptor 2; CAR, C-reactive protein/serum albumin ratio; ECOG, Eastern Cooperative Oncology Group performance status.

**Table 4 jpm-13-00923-t004:** Association between CAR and treatment response.

Tumor Response Data	Low-CAR (*n* = 27)	High-CAR (*n* = 37)	*p*-Value
Best response			
Complete response (CR)	2 (7.4)	0 (0.0)	0.18
Partial response (PR)	4 (14.8)	2 (5.4)	
Stable disease (SD)	8 (29.6)	8 (21.6)	
Progressive disease (PD)	12 (44.4)	25 (67.6)	
Not evaluated (NE)	1 (3.7)	2 (5.4)	
Overall response rate (ORR)	6 (22.2)	2 (5.4)	0.07
Disease control rate (DCR)	14 (51.9)	10 (27.0)	0.06

CAR, C-reactive protein/serum albumin ratio; ORR, CR+PR; DCR, CR+PR+SD.

**Table 5 jpm-13-00923-t005:** Comparison of any grade of adverse events between the low- and high-CAR groups. The bold data indicate statistically significant (*p* < 0.05).

	Low-CAR (*n* = 27)	High-CAR (*n* = 37)	*p*-Value
Total adverse events	22 (81.5)	29 (78.4)	1.000
Anorexia	9 (33.3)	17 (45.9)	0.440
Decreased neutrophil count	14 (51.9)	7 (18.9)	**0.008**
Fatigue	0 (0.0)	9 (24.3)	**0.008**
Diarrhea	1 (3.7)	3 (8.1)	0.632
Decreased platelet count	1 (3.7)	1 (2.7)	1.000
Anemia	0 (0.0)	1 (2.7)	1.000
Biliary tract infection	0 (0.0)	1 (2.7)	1.000
Febrile neutropenia	1 (3.7)	0 (0.0)	0.422
Hyponatremia	1 (3.7)	0 (0.0)	0.422
Myalgia	0 (0.0)	1 (2.7)	1.000
Peripheral sensory neuropathy	0 (0.0)	1 (2.7)	1.000

CAR: C-reactive protein/serum albumin ratio.

## Data Availability

The data presented in this study are available on request from the corresponding author.

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
