# Peer review of "Clinical Effect of the C-Reactive Protein to Serum Albumin Ratio in Patients with Metastatic Gastric or Gastroesophageal Junction Cancer Treated with Trifluridine/Tipiracil"

_jpm, 2023, doi:10.3390/jpm13060923_

Round 1

Reviewer 1 Report (New Reviewer)

The study evaluated the clinical effect of CAR in patients with metastatic gastric or EGJ cancer treated with FTD/TPI and concluded that high-CAR was an independent prognostic factor for OS and PFS.

However, in this study, I have some problems.

At first, the author categorized into two groups according to the median the cutoff of 0169.

Is this value justified among the small number of 64 patients? How can we judge high or low in actual clinical settings?

The title of this manuscript is in patients metastatic gastric or EGJ cancer treated with FTD/TPI. However, I think that the cohort is patients receiving third- or later-line chemotherapy rather than the patients treated with FTD/TPI. More than half of the patients in High CAR group can only use FTD/TPI once. Is FTD/TPI related to clinical outcome?

I can't believe that there are 2 cases of CR in Low CAR group. Do these two special cases affect the results?

What about the impact on the percentage of people who used ICI and the results?

Author Response

AUTHORS’ RESPONSES TO REVIEWERS’ COMMENTS

We thank the reviewers for their constructive critique to improve the manuscript. We have attempted to address the issues raised and to respond to all comments. The revisions are highlighted in red font in the revised manuscript. Please, find next a detailed, point-by-point response to the reviewers’ comments. We hope that our revisions will meet the reviewers’ expectations.

REVIEWER 1

The study evaluated the clinical effect of CAR in patients with metastatic gastric or EGJ cancer treated with FTD/TPI and concluded that high-CAR was an independent prognostic factor for OS and PFS. However, in this study, I have some problems. 

Point 1: At first, the author categorized into two groups according to the median the cutoff of 0169.

Is this value justified among the small number of 64 patients? How can we judge high or low in actual clinical settings?

Response 1:

We thank you for your valuable comment. As you highlighted, the median cutoff is somewhat less convincing and has questionable clinical application. We have added a new ROC analysis to statistically determine the cutoff. Therefore, all figures and tables have been revised. It is still possible that a high CAR is an independent predictor of prognosis.

Changes: “Median OS and PFS were significantly poorer in the high-CAR group than in the low-CAR group (113 vs. 399 days; P<0.001 and 39 vs. 112 days; P<0.001, respectively).” (Lines 28–30)

“The cutoff value, defined using a receiver operating characteristic curve analysis for survival and death, was 0.09 (Figure 1). The patients were categorized into the high and low CAR group.” (Lines 93–95)

“Figure 1. Definition of the cutoff value using receiver operating characteristic curve analysis on survival in CAR. AUC, area under the curve; CAR, C-reactive protein/serum albumin ratio.” (Lines 99–100)

“There were 37 and 27 patients in the high and low CAR groups, respectively, using the cutoff point for the CAR.” (Lines 116–117)

“The high-CAR group had a significantly worse Eastern Cooperative Oncology Group performance status (P=0.022), a higher prevalence of patients receiving only a single course of FTD/TPI (P=0.018), and a higher rate of patients who could not be administered any post-FTD/TPI therapy (P=0.004) than the low-CAR group. However, no significant relationships were observed among age (P=0.797), sex (P=0.600), histological type (P=0.621), human epidermal growth factor receptor 2 status (P=1.000), macroscopic classification (P= 0.058), primary tumor site (P=1.000), peritoneal metastasis (P=0.129), hematogenous metastasis (P=0.613), lymphatic metastasis (P=0.802), number of metastatic sites (P=0.454), number of previous regimens (P=0.551), and previous immune checkpoint inhibitor administered (P=1.000) between the high- and low-CAR groups (Table 1).” (Lines 120–129)

“Table 1. Clinicopathological data between low- and high-CAR groups” (Line 130)

“The median OS of patients with high CAR was significantly poorer than that of those with low CAR (113 days vs. 399 days; P<0.001) (Figure 2). Additionally, the median PFS of patients with high CAR was significantly poorer than that of those with low CAR (39 days vs. 112 days; P<0.001) (Figure 3).” (Lines 135–138)

“Figure 2. Kaplan–Meier curve for OS between the high- and low-CAR groups. The median OS of patients with high CAR was significantly shorter than that of those with low CAR (113 days vs. 399 days; P<0.001). OS, overall survival; CAR, C-reactive protein-to-serum albumin ratio” (Lines 140–142)

“Figure 3. Kaplan–Meier curve for PFS between the high- and low-CAR groups. The median PFS of patients with high CAR was significantly poorer than that of those with low CAR (39 vs. 112 days; P<0.001). CAR, C-reactive protein-to-serum albumin ratio; PFS, progression-free survival” (Lines 144–146)

“Table 2 shows the results of univariate and multivariate analyses of OS in patients with GC who underwent FTD/TPI. In univariate analysis, ECOG PS (HR, 1.83; 95% CI, 1.02–3.29; P=0.04), peritoneal metastasis (HR, 2.57; 95% CI, 1.37–4.82; P=0.003), and CAR (HR, 4.48; 95% CI, 2.24–8.97; P<0.001) were prognostic factors for OS. In multivariate analyses, CAR (HR, 3.56; 95% CI, 1.67–7.55; P<0.001) was an independent prognostic factor for OS (Table 2). Table 3 shows the results of univariate and multivariate analyses of PFS in patients with GC who underwent FTD/TPI. In univariate analysis, peritoneal metastasis (HR, 1.78; 95% CI, 1.04–3.05; P=0.04) and CAR (HR, 2.67; 95% CI, 1.56–4.66; P<0.001) were prognostic factors for PFS. However, CAR (HR, 2.42; 95% CI, 1.33–4.41; P=0.004) was an independent prognostic factor for PFS in multivariate analyses (Table 3).” (Lines 148–157)

“Table 2. Univariate and multivariate analyses of clinicopathological factors and CAR for OS” (Line 158)

Table 3. Univariate and multivariate analyses of clinicopathological factors and CAR for progression-free survival” (Lines 162–163)

“The rates of complete response were 0.0% (n = 0) and 7.4% (n = 2), partial response was 5.4% (n = 2) and 14.8% (n = 4), stable disease was 21.6% (n = 8) and 29.6% (n = 8), progressive disease was 67.6% (n = 25) and 44.4% (n=12), and not evaluated were 5.4% (n = 2) and 3.7% (n = 1) in the high- and low-CAR groups, respectively. The overall response rate (ORR) was 5.4% (n = 2) and 22.2% (n = 6) in the high- and low-CAR groups, respectively, which was superior to the high-CAR group but not significantly different (Table 4). The disease control rates (DCRs) were 27.0% (n = 10) and 51.9% (n = 14) in the high- and low-CAR groups, respectively, which were superior to the high-CAR group but not significantly different (Table 4).” (Lines 167–175)

“Table 4. Association between CAR and treatment response” (Line 176)

“The high-CAR group had a significantly lower and higher incidence of neutropenia (P=0.008) and fatigue (P=0.008), respectively, than the low-CAR group (Table 5). However, no significant relationships were observed with total adverse events (P=1.000), anorexia (P=0.440), diarrhea (P=0.632), decreased platelet count (P=1.000), anemia (P=1.000), biliary tract infection (P=1.000), febrile neutropenia (P=0.422), hyponatremia (P=0.422), myalgia (P=1.000), and peripheral sensory neuropathy (P=1.000) between the high- and low-CAR groups.” (Lines 180–186)

“Table 5. Comparison of any grade of adverse events between the low- and high-CAR groups” (Line 188)

Point 2: The title of this manuscript is in patients metastatic gastric or EGJ cancer treated with FTD/TPI. However, I think that the cohort is patients receiving third- or later-line chemotherapy rather than the patients treated with FTD/TPI. More than half of the patients in High CAR group can only use FTD/TPI once. Is FTD/TPI related to clinical outcome?

Response 2:

We thank you for your insightful comment. As you noted, the high CAR group had significantly fewer FTD/TPI treatments than the Low CAR group. This may be because the high CAR group did not fully benefit from FTD/TPI treatment due to disease status and side effects, resulting in a poorer prognosis. Therefore, we have added the following text to the discussion.

Changes: “In fact, in this study, the high-CAR group had a higher prevalence of patients receiving only a single course of FTD/TPI than the low-CAR group. This suggests that the high CAR group has not fully benefited from FTD/TPI treatment regarding disease status and adverse effects. which may contribute to poor prognosis.” (Lines 277–281)

Point 3: I can't believe that there are 2 cases of CR in Low CAR group. Do these two special cases affect the results?

Response 3: We thank you for your comment. Survival analysis of all but two CR cases showed that the high CAR group had a significantly worse prognosis regarding OS and PFS than the Low CAR group (113 days vs. 399 days, P<0.001 and 39 days vs. 97 days, P=0.003).

Point 4: What about the impact on the percentage of people who used ICI and the results?

Response 4: We thank you for your comment. Clinicopathologic analysis was added for using ICI, with no significant difference between the high CAR and low CAR groups in using ICI.

Changes: “Table 1. Clinicopathological data between low- and high-CAR groups” (Line 130)

Reviewer 2 Report (Previous Reviewer 2)

1. Why is the age of patients divided by 65 years old?

2. Please add the description of the collected method of blood samples in the M&M.

Minor editing of English language required.

Author Response

AUTHORS’ RESPONSES TO REVIEWERS’ COMMENTS

We thank the reviewers for their constructive critique to improve the manuscript. We have attempted to address the issues raised and to respond to all comments. The revisions are highlighted in red font in the revised manuscript. Please, find next a detailed, point-by-point response to the reviewers’ comments. We hope that our revisions will meet the reviewers’ expectations.

REVIEWER 2

Point 1: Why is the age of patients divided by 65 years old?

Response 1: We thank you for your comment. The United Nations World Health Organization (WHO) defines the elderly as people aged 65 years or older; therefore, we used age 65 as the cutoff point for this study.

Point 2: Please add the description of the collected method of blood samples in the M&M.

Response 2: We thank you for your valuable suggestion. We have corrected the M&M as you indicated.

“Clinical blood samples from patients were typically transported directly to the blood draw room for immediate analysis. Blood samples were used to measure blood counts, biochemistry, coagulation factors, and tumor markers.” (Lines 88–90)

Point 3: Minor editing of English language required.

Response 3: We thank you for your valuable comment. The manuscript has been revised, edited in English, and submitted.

Round 2

Reviewer 1 Report (New Reviewer)

The author responds appropriately to the reviewer’s comments. I think that readers of JCM will be interested in this study.

Reviewer 2 Report (Previous Reviewer 2)

The authors have corrected this manuscript according to the reviewers' suggestions. Therefore, this manuscript will be considered to accept. 

This manuscript is a resubmission of an earlier submission. The following is a list of the peer review reports and author responses from that submission.

Round 1

Reviewer 1 Report

The manuscript describes a retrospective analysis on 64 patients with advanced gastric or gastroesophageal junction (GEJ) adenocarcinomas who were treated with TAS102 (Trifluridine/Tipiracil) in the palliative setting in a late treatment line. The prognostic role of the ratio between C-reactive protein (CRP) and serum albumin (CAR) is studied. A prognostic role of CAR for PFS and OS is demonstrated both in univariate and multivariate analysis.

The manuscript is overall well written, and the statistical methods are adequate. There are several points that should be addressed:

By definition, a prognostic biomarker is indicative for the outcome irrespective of the therapy given. This should be clearly discussed to avoid that the impression is created, the marker specifically predicts the efficiency of TAS102.

The statement, that prognostic biomarkers in gastric or GEJ adenocarcinomas to be limited is not correct. Regarding the parameters used in this analysis (CRP and albumin) as an example, the Glasgow Prognostic Score (GPS) which includes the same parameters and other inflammation related parameters have been shown to be prognostic in a meta-analysis for gastric cancer (Mao et al. C-reactive protein/albumin and neutrophil/lymphocyte ratios and their combination predict overall survival in patients with gastric cancer. Oncol Lett. 2017;14(6):7417-24).

It would be interesting at this point, if CAR used in this analysis differs from GPS regarding the prognostic power.

Patients with high CAR had less hematotoxicity according to the data presented. How did this correlate with the doses of TAS102 applied? Could it be that patients with high CAR received less chemotherapy?

The discussion on the influence of inflammation on the expression of thymidine phosphorylase is not supported by data from the cohort described. But this discussion again creates the impression, that CAR might specifically predict the efficiency of TAS102.

Overall, the cohort is quite small (n=64), and the novelty is limited. A predictive role for CAR as a biomarker for TAS102 efficiency can only be demonstrated in a cohort that includes both TAS102 treated and untreated patients like in the pivotal study by Shitara et al. Trifluridine/tipiracil versus placebo in patients with heavily pretreated metastatic gastric cancer (TAGS): a randomised, double-blind, placebo-controlled, phase 3 trial. The Lancet Oncology. 2018;19(11):1437-48. Such an analysis would be very interesting.

Reviewer 2 Report

This manuscript 'Clinical effect of the C-reactive protein to serum albumin ratio in patients with metastatic gastric or gastroesophageal junction cancer treated with trifluridine/tipiracil' is a clinical study. In this manuscript, some parts should be clearly described following as:

1. In M&M, please describe the assay methods of C-reactive protein and serum albumin.

2. line 93: CAR = C-reactive protein (mg/dl)/Alb (g/dl). Please modify it to CAR = C-reactive protein (mg/dl)/serum albumin (g/dl).

3. Table 3, 4, and 5: p-Value or P-Value.